# Antcins from *Antrodia cinnamomea* and *Antrodia salmonea* Inhibit Angiotensin-Converting Enzyme 2 (ACE2) in Epithelial Cells: Can Be Potential Candidates for the Development of SARS-CoV-2 Prophylactic Agents

**DOI:** 10.3390/plants10081736

**Published:** 2021-08-23

**Authors:** K. J. Senthil Kumar, M. Gokila Vani, Han-Wen Hsieh, Chin-Chung Lin, Sheng-Yang Wang

**Affiliations:** 1Bachelor Program of Biotechnology, National Chung Hsing University, Taichung 402, Taiwan; zenkumar@dragon.nchu.edu.tw; 2Department of Forestry, National Chung Hsing University, Taichung 402, Taiwan; mgvani2009@gmail.com; 3Taiwan Leader Biotech Company, Taipei 103, Taiwan; ck_hsieh@twleaderlife.com (H.-W.H.); johnson@twleaderlifw.com (C.-C.L.); 4Agricultural Biotechnology Research Center, Academia Sinica, Taipei 11529, Taiwan

**Keywords:** SARS-CoV-2, COVID-19, ACE2, antcin, *Antrodia cinnamomea*, *Antrodia salmonea*

## Abstract

Antcins are newly identified steroid-like compounds from Taiwan’s endemic medicinal mushrooms *Antrodia cinnamomea* and *Antrodia salmonea*. Scientific studies of the past two decades confirmed that antcins have various pharmacological activities, including potent anti-oxidant and anti-inflammatory effects. The severe acute respiratory syndrome coronavirus 2 (SARS-CoV-2) caused the coronavirus disease-2019 (COVID-19) pandemic and is characterized as a significant threat to global public health. It was recently identified that SARS-CoV-2 required angiotensin converting enzyme 2 (ACE2), a receptor which supports host cell entry and disease onset. Here, we report a novel function of antcins, in which antcins exhibit inhibitory effects on ACE2. Compared to the untreated control group, treatment with various antcins (antcin-A, antcin-B, antcin-C, antcin-H, antcin-I, and antcin-M) significantly inhibited ACE2 activity in cultured human epithelial cells. Indeed, among the investigated antcins, antcin-A, antcin-B, antcin-C, and antcin-I showed a pronounceable inhibition against ACE2. These findings suggest that antcins could be novel anti-ACE2 agents to prevent SARS-CoV-2 host cell entry and the following disease onset.

## 1. Introduction 

Acute respiratory syndrome coronavirus 2 (SARS-CoV-2) is a causative pathogen responsible for the ongoing pandemic coronavirus disease 2019 (COVID-19) that affected more than 222 countries with over 191 million cases and claimed 4.1 million lives as of today (https://www.worldometers.info/coronavirus/ and accessed on 21 July 2021). Currently, available pharmaceuticals face several limitations for treating COVID-19. The major challenges ranged from safety, side-effects, bio-availability, and poor efficacy, which accelerated the search for better treatment modalities. Since COVID-19 was characterized as severe pneumonia leading to persistent complications, including lung damage and multi-organ failure, preventive strategies are the best option [1]. In this pursuit, the development of vaccines and prophylactic agents are the primary targets. Within one year of the pandemic, several vaccines have been developed to combat the viral spread. Some of these vaccines are rolling out globally to minimize the infection and disease severity, and some of these vaccines are currently under clinical trials. 

Additionally, the number of anti-viral drugs have been prescribed to treat severe COVID-19 patients. One of the repurposed drugs is remdesivir, used under emergency medical use to treat hospitalized COVID-19 patients. Despite, insufficient evidence to disclose the benefits and risks of remdesivir, the World Health Organization (WHO) recommended suspending the use of remdesivir for COVID-19 patients [2]. Although, the number of COVID-19 positive cases and deaths are continuously increasing due to the unavailability of effective drugs to prevent or slow down the infection process. Thus, the search for new drugs to prevent or block the infection needs to be urgently intensified. One of the potential preventive targets of anti-COVID-19 drugs is angiotensin-converting enzyme 2 (ACE2). The SARS-CoV-2 spike (S) protein, a type-I membrane-bound glycoprotein projection from viral envelope, can directly bind to ACE2, a membrane-associated enzyme in host cells [3]. Recently, it was identified that the high affinity of SARS-CoV-2 to ACE2 is a crucial event in human-to-human transmission [4]. Therefore, diminishing the protein expression of ACE2 and TMPRSS2 in host cells can be an effective strategy for the prophylaxis and treatment of SARS-CoV-2 infection.

Medicinal herbs play a vital role in disease prevention and treatment, especially in dealing with easily transmitted diseases in the past [5]. Indeed, phytocompounds are widely recognized as promising prophylactic and therapeutic candidates against various viral infections [6]. ACE2 has been identified as a host cell surface receptor for the entry of SARS-CoV-2 in humans, which is considered a prominent target for development of new therapies [7]. Several phytocompounds are reported to possess ACE2 inhibitory activity and are extensively investigated with in silico, pre-clinical, and clinical models [8,9,10]. More than 300 plant extracts have reported ACE2 inhibition activity, some of them are common herbs, including cinnamon, pepper, olive, black nightshade, hawthorn, passion fruit, and grapes. Additionally, ACE2 inhibitors from plant extracts belongs to several phytochemical classes, including alkaloids, flavonols, flavonones, terpenes, limonoids, lignans, terpenoids, tannins, phenolic acids, and fatty acids [11].

*A. cinnamomea* and *A. salmonea (Fomitopsidaceae) are endemic fungus of Taiwan, which are traditionally used as a health promoting supplement by the indigenous people* [12]. *A. cinnamomea* and *A. salmonea* were reported to possess various pharmacological activities, including anti-microbial, anti-oxidant, anti-inflammation, anti-cancer, anti-diabetic, anti-aging, immunomodulatory, hepatoprotective, cardioprotective, and neuroprotective effects [13,14]. The broad-spectrum of pharmacological activities of *A. cinnamomea* and *A. salmonea* are characterized to be the result of a pool of various and unique bioactive compounds that include terpenoids, benzenoids, benzoquinone, succinic acid, and maleic acid derivatives, nucleic acids, polysaccharides, lignans, and steroids. A study by metabolic profiling and comparison of bioactivities of *A. cinnamomea* and *A. salmonea* revealed that the chemical composition and bioactivities were varied. For example, 3,4,5-trimethoxybenzaldehyde, δ-guaiene, isolongifolene, 1-octen-3-ol, 4-terpinenol, α-guaiene, and *p*-cymene were found to be the most abundant compounds in *A. cinnamomea*, while α-cedrene, 1-octen-3-ol, D-limonene, cadinadiene, germacrene D, isolongifolene, and α-muurolene were the main compounds of *A. salmonea*. In addition, (*R*,*S*)-antcin-B and (*R*,*S*)-antcin-C were the most abundant compound in *A. cinnamomea* and *A. salmonea* fruiting bodies, respectively. Indeed, antcin-M and methyl antcinate-K, were only found in *A. salmonea*; therefore, antcin-M and methyl antcinate-K can be key compounds to segregate *A. cinnamomea* and *A. salmonea* fruiting bodies [15]. Antcins are unique steroid-like compounds isolated *A. cinnamomea* and *A. salmonea*. At present, twelve antcins, *i.e.,* antcin A, B, C, D, E, F, G, H, I, K, M, and N, along with twelve derivatives/epimers (25*R*/*S*-antcin A, B, C, H, I, and K) and seven analogs (methyl antcinate A, B, G, H, K, L, and N) were identified. Several studies have demonstrated that antcins possessed anti-cancer, anti-inflammation, anti-oxidant, anti-diabetic, anti-aging, immunomodulation, hepatoprotection, and hypolipidemic activities [14]. However, anti-viral and prophylactic effect of viral infection was poorly studied. Recently, several natural products have been screened as potential preventive or therapeutic agents for COVID-19 [16,17,18,19]. However, most studies have used in silico simulation models to identify potential agents [9,17], while few studies were conducted with in vitro [20,21] or in vivo models [22]. In this study, we investigated the ACE2 inhibitory effect of various antcins, including antcin-A, antcin-B, antcin-C, antcin-H, antcin-I, and antcin-M (Figure 1) in cultured human colon epithelial cells in vitro. 

## 2. Results and Discussion

Studies have identified abundant ACE2 expression in both transformed and primary epithelial cell lines, including renal (ACHN, HEK293, and RPTECs), colon (CACO-2 and HT-29), and cardiovascular (HUVECs, HCAECs, SVSMCs, and cardio fibroblasts) [23]. In addition, a recent study identified ACE2 overexpression in various human organs, including lungs, kidneys, esophagus, colon, heart, brain, prostate, liver, tongue, and other organs. Indeed, high endogenous expression of ACE2 has been found in the colon, gallbladder, and heart muscle tissues [24]. These studies were pointing out that gastrointestinal tissues and colon cell lines can be a platform to study the ACE2 activity [23,24,25]. Therefore, in this study, we subjected HT-29, a colon adenocarcinoma cell line, to investigate the ACE2 inhibitory effects of antcins. Before investigating the ACE2 inhibitory effect of antcins, we determined the cytotoxicity of various antcins on HT-29 cells. HT-29 cells were incubated with increasing concentrations (5–40 M) of antcins, including antcin-A, antcin-B, antcin-C, antcin-H, antcin-I, antcin-K, and antcin-M for 48 h, and then the cell viability was measured by 3-(4,5-dimethyl-thiazol-2-yl)-2,5-diphenyl tetrazolium bromide (MTT) colorimetric assay. Based on the results, we found that antcins displayed differential cytotoxicity to HT-29 cells (Figure 2). Antcin-A, antcin-C, and antcin-M did not exhibit cytotoxicity to HT-29 cells up to a dose of 40 M, while high concentrations of antcin-H and antcin-I displayed a reduction in cell viability; however, this reduction was not statistically significant. Incubation of HT-29 cells with 40 M antcin-B resulted from significant reduction in cell viability (85.48%), whereas cells were survived up to a dose of 20 M. In this antcins series, antcin-K showed significant cytotoxicity to HT-29 cells, which reduced the cell viability to 84.72%, 65.84%, and 40.36% by 10, 20, and 40 M of antcin-K. Based on the MTT assay, we selected the optimum non-cytotoxic concentration of antcins, 40, 20, 40, 20, 20, and 20 M of antcin-A, antcin-B, antcin-C, antcin-H, antcin-I, and antcin-M to use for our further experiments. Due to the cytotoxicity, antcin-K was suspended from the list for further investigations.

Previous studies based on *in silico* modeling and virtual screening demonstrated that triterpenoids, which has similar polarity and common origin with antcins exhibited their anti-SARS-CoV-2 activity through diverse mechanisms. For example, Vardhan and Sahoo [26] reported that four triterpenoids, namely ursolic acid, corosolic acid, maslinic acid, and glycyrrhizic acid, exhibited strong affinity with primary therapeutic targets of SARS-CoV-2, including main protease (3CLpro), papain-like protease (PLpro), spike glycoprotein-receptor binding domain (SGp-RBD), and RNA dependent RNA polymerase (RdRp). Additionally, they revealed that six triterpenoids, ursolic acid, maslinic acid, glycyrrhizic acid, azadiradionolide, epoxyazadiradione, and gedunin were found to bind at the catalytic site of ACE2. Another study by Carino et al. [27] revealed that betulin, betulinic acid, oleanolic acid, and glycyrrhetinic acid could bind at the RBD of the SARS-CoV-2 S protein. Additionally, oleanolic acid and glycyrrhetinic acid have been proven effective in reducing the spike RBD’s adhesion to its ACE2 consensus in vitro. The structural optimization study with 3-*O*-chacotriosyl oleanane-type triterpenoid inhibits SARS-CoV-2 virus entry via binding to SARS-CoV-2 glycoprotein (S) [28]. In addition, structurally similar synthetic corticosteroids such as dexamethasone exert anti-inflammatory effects and are prescribed medicines for critically ill COVID-19 patients. However, the WHO advised restricting the long-term use or high-dose corticosteroids due to steroid-induced osteonecrosis of the femoral head. Whether corticosteroids should be used in COVID-19 is, therefore, uncertain. Indeed, our previous structural activity relationship study revealed that antcins are structurally similar to glucocorticosteroids, and antcin-A exhibited a strong anti-inflammatory effect on human lung epithelial cells [29]. Therefore, we sought to examine whether antcins can inhibit ACE2 activity in human epithelial cells. HT-29 cells were incubated with the maximum non-cytotoxic concentration of antcins for 48 h. The total levels of both cytoplasmic and membrane-bound ACE2 were determined by a commercially available ELISA kit. We found that all the treated antcins, except antcin-M, exhibited significant inhibition on ACE2 activity (Figure 3). Indeed, antcin-A, antcin-B, antcin-C, and antcin-I strongly reduced human ACE2 levels in HT-29 cells from 11.23 ng/mL (control) to 4.39 ng/mL, 4.22 ng/mL, and 4.19 ng/mL, respectively, while a moderate reduction was observed in antcin-H treated cells (5.91 ng/mL). Recently, we reported that geranium essential oil and its major bioactive compound citronellal showed potent inhibition against human ACE2 [30]. Therefore, citronellal (50 M) was used as a positive drug control to compare the ACE2 inhibitory effect of antcins. Based on the result, antcins are highly competitive with citronellal. 

To further confirm the ACE2 inhibitory effect, we examined the protein expression levels of ACE2 by Western blotting. Like the ELISA result, the ACE2 protein expression level was significantly reduced by all the treated antcins (Figure 4A). However, antcin-H-mediated reduction in ACE2 protein level was not statistically significant when compared with a control group. We recently reported that essential oils and their active components significantly down-regulated ACE2 and TMPRSS2 at the transcriptional level [30]. Therefore, we next examined whether the antcin-mediated reduction in ACE2 protein level was associated with down-regulation of ACE2 mRNA level in HT-29 cells. Interestingly, all the tested antcins failed to inhibit either ACE2 (Figure 4B) or TMPRSS2 (Figure 4C) mRNA expression in HT-29 cells, suggesting that antcin-induced reduction in ACE2 was not caused by down-regulation at the mRNA level. Recently, it was identified that some chemicals induce ACE2 protein degradation without affecting its bio-synthesis: for example, benzo(a)pyrene-modulated ACE2 protein stability in human normal lung epithelial 16HBE cells via enhanced E3-ubiquitination [31]. Therefore, it is reasonable to the hypothesis that antcin-induced reduction in ACE2 was most likely related to ACE2 protein stability or degradation by proteasome pathway rather than transcriptional inhibition. However, these findings are in preliminary stage; therefore, at this point we cannot claim that antcins can be prophylactic or therapeutic agents for clinical use. We believe that further pharmacological, toxicological, and bioavailability studies may support our hypothesis that antcins can be a potential ACE2 inhibitors for prophylaxis of COVID-19. 

## 3. Materials and Methods

### 3.1. Chemicals and Reagents

Antcin-A, antcin-B, antcin-C, antcin-H, antcin-I, antcin-K, and antcin-M were isolated from either *A. cinnamomea* or *A. salmonea,* as described previously [15]. The purity of the antcins was above 99%, as confirmed by HPLC and FT-NMR analysis. McCoy’s medium, fetal bovine serum (FBS), sodium pyruvate, penicillin, and streptomycin were obtained from Invitrogen (Carlsbad, CA, USA). 3-(4,5-Dimethyl-thiazol-2-yl)-2,5-diphenyl tetrazolium bromide (MTT) and dimethylsulfoxide (DMSO) was purchased from Sigma-Aldrich. Antibody against ACE2 was obtained from Arigo Biolaboratories (Hsinchu, Taiwan). The antibody against GAPDH was obtained from Cell Signaling Technology Inc. All other chemicals were reagent grade or HPLC grade and were supplied by either Merck or Sigma-Aldrich.

### 3.2. Cell Culture and Sample Treatment

The human colorectal adenocarcinoma (HT-29) cell line was cultured and maintained in McCoy’s medium supplemented with 10% fetal bovine serum (FBS), 1% L-Glutamine, and 1% penicillin/streptomycin. Cells were incubated with increasing concentrations of antcins (5–40 M) for 48 h. All the antcins were dissolved in DMSO and an equal volume of DMSO (0.1%) was used as vehicle control. 

### 3.3. Cell Viability Assay

Cell viability was assessed by MTT colorimetric assay. Briefly, HT-29 cells were seeded into a 48-well culture plate with a density of 5 × 10^4^ cells/well. After 24 h incubation, cells were treated with increasing concentrations of antcins (5–40 M) for 48 h. The culture supernatant was removed, and 1 mg/mL of MTT in 0.1 mL of fresh medium was added. The MTT formazan crystals were dissolved in 0.4 mL of DMSO and the samples were measured at 570 nm (A_570_) using ELISA microplate reader (Bio-Tek Instruments, Winooski, VT, USA). The percentage of cell viability (%) was calculated as (A_570_ of treated cells/A_570_ of untreated cells) × 100. 

### 3.4. Determination of ACE2 Activity

Cellular ACE2 activity was measured using a commercially available human ACE2 ELISA kit (Elabscience Biotechnology Inc, Hubei, China). Briefly, HT-29 cells were seeded in a 10 cm cell culture dish with a density of 2 × 10^6^ cells/well. After 24 h, cells were treated with a selective dose (non-cytotoxic high concentration) of antcins for 48 h. Then, the cells were gently washed with pre-cooled phosphate-buffered saline (PBS), and the cells were dissociated using trypsin. The cell suspension was centrifuged for 5 min at 1000× *g*. The cell pellets were lysed with a mixture of radioimmunoprecipitation assay (RIPA) buffer (Thermo Fisher Scientific, Waltham, MA, USA) and trans-membrane protein extraction reagent (Fivephoton Biochemical, San Diego, CA, USA) at a ratio of 1:1, followed by centrifugation for 10 min at 1500× *g* at 4 °C. The protein concentration in the samples were determined by a Bio-Rad protein assay reagent (Bio-Rad Laboratories, Hercules, CA, USA). Equal amounts of protein samples (100 µg) were assayed according to the manufacturer’s protocol. The optical density (OD value) of each sample was determined at 450 nm (A_450_) using an ELISA microplate reader (Bio-Tek Instruments). The percentage of ACE2 activity (%) was calculated as (A_450_ of treated cells/A_450_ of untreated cells) × 100.

### 3.5. Determination of ACE2 Protein

HT-29 cells were seeded in a 10 cm dish at a density of 2 × 10^6^ cells/dish. After 24 h, the cells were incubated with various antcins (20–40 M) for 48 h. Cellular and membrane-bound proteins were isolated by a mixture of RIPA lysis buffer and trans-membrane protein extraction reagent. Protein concentration was determined by a Bio-Rad protein assay reagent. Equal amounts of denatured protein samples (60 µg) were separated by 10% SDS-PAGE, and the separated proteins were transferred onto polyvinylidene difluoride (PVDF) membrane overnight. Then, the membranes were blocked with 5% non-fat dried milk for 30 min, followed by incubation with ACE2 or GAPDH antibodies overnight, and then incubated with either horseradish peroxidase-conjugated goat anti-rabbit or anti-mouse antibodies for 1 h. Immunoblots were developed with enhanced chemiluminescence (ECL) reagents (Advansta Inc., San Jose, CA, USA), images were captured by ChemiDoc XRS^+^ docking system, and the protein bands were quantified by Imagelab software (Bio-Rad Laboratories, Hercules, CA, USA).

### 3.6. Quantitative Real-Time PCR

Total RNA was extracted by a total RNA purification kit (GeneMark, New Taipei City, Taiwan). The RNA concentration was quantified using a NanoVue Plus spectrophotometer (GE Health Care Life Sciences, Chicago, IL, USA). Quantitative PCR (qPCR) was performed on a real-time PCR detection system and software (Applied Biosystems, Foster City, CA, USA). First-strand complementary DNA (cDNA) was generated by SuperScript III reverse transcriptase kit (Invitrogen). Quantification of mRNA expression for genes of interest was performed by qPCR reactions with an equal volume of cDNA, forward and reverse primers (10 µM), and Power SYBR Green Master Mix (Applied Biosystems). The sequence of the PCR primers was as follows: ACE2: forward 5′-GCTGCTCAGTCCACCATTGAG-3′, reverse 5′-GCTTCGTGGTTAAACTTGTCCAA-3′; TMPRSS2: forward 5′-AATCGGTGTGTTCGCCTCTAC-3′, reverse 5′-GCGGCTGTCACGATCC-3′; GAPDH: forward 5′-TCCTGGTATGACAACGAAT-3′, reverse 5′-GGTCTCTCTCTTCCTCTTG-3′ [30]. The copy number of each transcript was calculated as the relative copy number normalized by the GAPDH copy number.

### 3.7. Statistical Analysis

Data are expressed as mean ± SD. All data were analyzed using the statistical software Graphpad Prism version 6.0 for Windows (GraphPad Software, San Diego, CA, USA). Statistical analysis was performed using one-way ANOVA followed by Dunnett’s test for multiple comparisons. *p*-values of less than 0.05 *, 0.01 **, 0.001 ***, and 0.0001 **** were considered statistically significant for the sample treatment group *vs*. the control group. 

## Figures and Tables

**Figure 1 plants-10-01736-f001:**
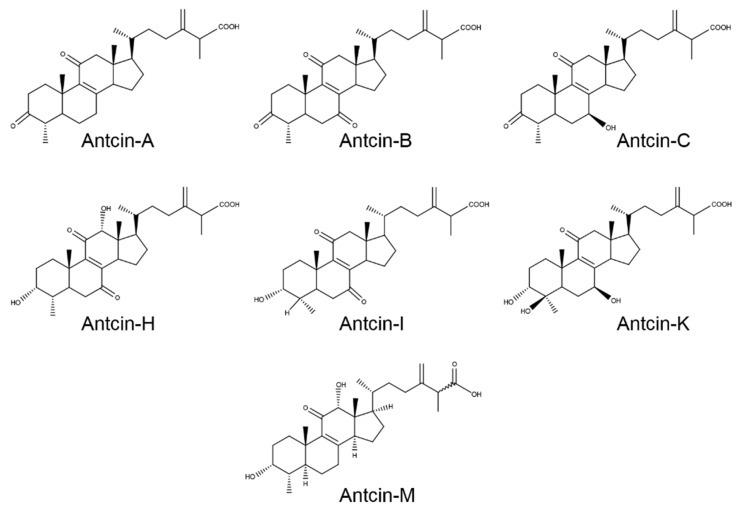
Chemical structure of antcins.

**Figure 2 plants-10-01736-f002:**
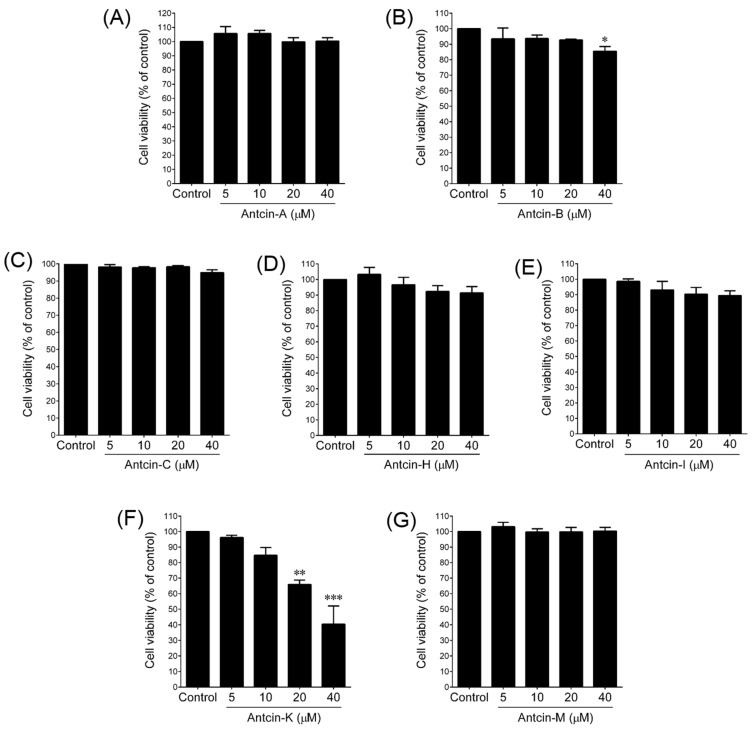
Cytotoxic effects of antcins on HT-29 cells. (**A**–**G**) HT-29 cells were incubated with increasing concentrations of antcins (5–40 M) for 48 h. The cell viability was determined by the MTT colorimetric assay as described in materials and methods. Values represent the mean ± SD of three independent experiments. *p*-values of less than 0.05 *, 0.01 **, and 0.001 *** were considered statistically significant for the antcins treatment group *vs*. the control group.

**Figure 3 plants-10-01736-f003:**
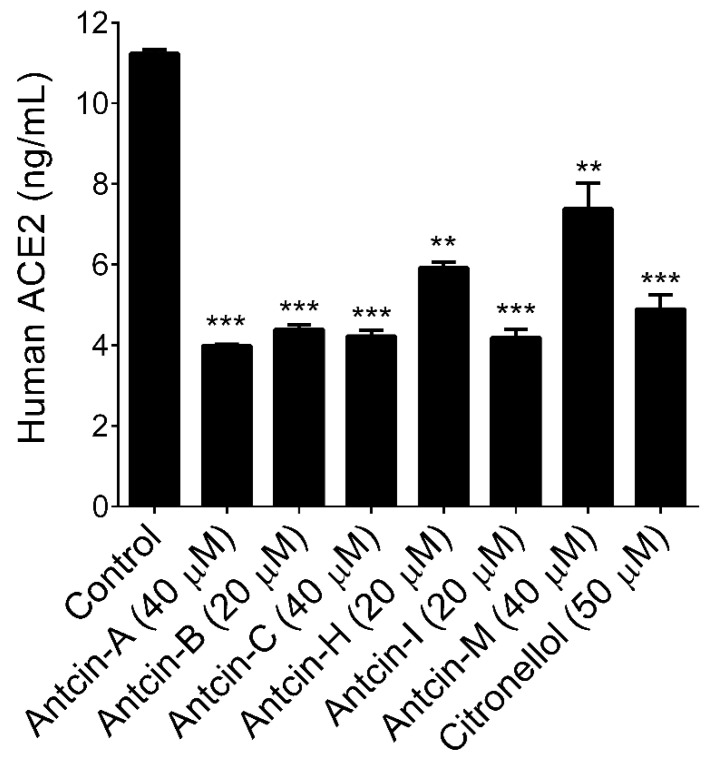
Antcins reduced human ACE2 levels in HT-29 cells. HT-29 cells were incubated with the indicated concentration of antcins for 48 h. Cell lysates were subjected to determine ACE2 levels using a commercially available ELISA kit. Values represent the mean ± SD of three independent experiments. *p*-values of less than 0.01 **, and 0.001 *** were considered statistically significant for the antcins treatment group vs. the control group.

**Figure 4 plants-10-01736-f004:**
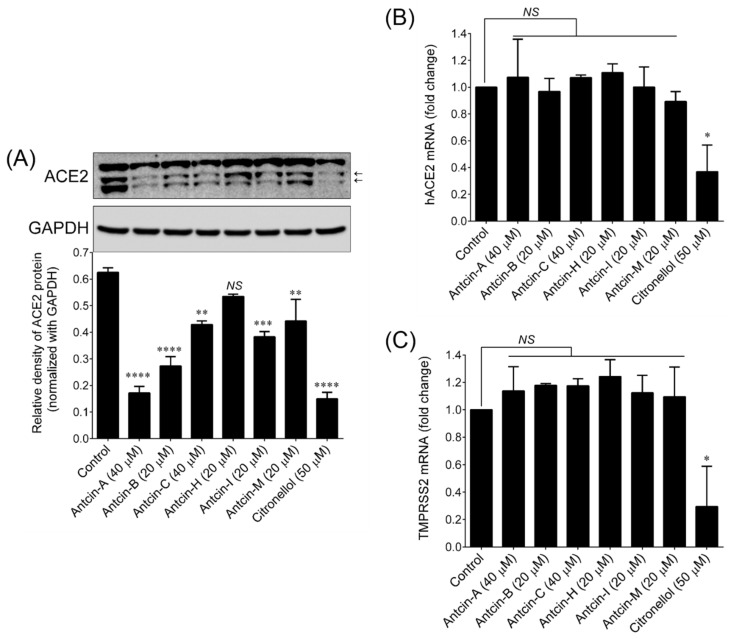
Antcins inhibited ACE2 protein expression in HT-29 cells. (**A**) HT-29 cells were incubated with the indicated concentration of antcins for 48 h. Protein expression levels of ACE2 was determined by immunoblotting. Glyceraldehyde 3-phosphate dehydrogenase (GAPDH) was used as an internal loading control. The relative density of one representative experiment is shown, where ACE2 signal was normalized with GAPDH signal. (**B**,**C**) Relative expression of ACE2 and TMPRSS2 mRNAs in HT-29 cells. Total RNA was extracted from cells treated with indicated concentration of antcins for 48 h. The transcription levels of ACE2 and TMPRSS2 were quantified by qPCR and a representative experiment is shown. The Δ^ct^ values of ACE2 and TMPRSS2 mRNAs were normalized to GAPDH mRNA. Values represent the mean ± SD of two independent experiments. *p*-values of less than 0.05 *, 0.01 **, 0.001 ***, and 0.0001 **** were considered statistically significant for the antcins treatment group vs. the control group.

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
