# Peer review of "Antcins from Antrodia cinnamomea and Antrodia salmonea Inhibit Angiotensin-Converting Enzyme 2 (ACE2) in Epithelial Cells: Can Be Potential Candidates for the Development of SARS-CoV-2 Prophylactic Agents"

_plants, 2021, doi:10.3390/plants10081736_

Round 1

Reviewer 1 Report

The study by K.J. Senthil Kumar et al. entitled “Antcins from Antrodia cinnamomea and Antrodia salmonea as 2 Potential Angiotensin-Converting Enzyme 2 (ACE2) Inhibitors 3 for the Prophylaxis of SARS-CoV-2 Infection” has been reviewed. The study aimed to evaluate the ACE2 inhibitory effect on human colon epithelial cells of various antcins (antcin-A, antcin-B, antcin-C, antcin-H, antcin-I, and antcin-M) isolated from the fungus, Antrodia cinnamomea and Antrodia salmomea endemic of Taiwan.

The manuscript is potentially interesting, the rationale is clear, figures and legends are exhaustive and the flow is well structured. Some suggestions are listed below.

The title is too speculative for the data reported in a short report, many studies are still needed to verify the efficacy and safety of antcins in the prophylaxis of SARS-CoV-2 infection. Please rephrase it

In the introduction, please reduce the background on SARS-CoV-2 and introduce more detailed information on the two species of fungi and metabolites studied.

Why the authors did not use nasal or lung epithelial cells in consideration of the principle of prophylaxis? Lane 105 Add the appropriate reference “Studies were pointing out that gastrointestinal tissues and 104 colon cell lines can be a platform to study the ACE2 activity”

Please also concisely describe the extraction method and all the steps for the separation of individual metabolites.

I would urge the authors to accommodate the suggestions reported above, to further improve the quality of the manuscript.

Reviewer 2 Report

The study is generally interesting due to the new information on the potential activities of phytochemicals belonging to the class of steroids. However, it was performed (as majority of such research) on cultured cells, and therefore, in my opinion, no convincing final conclusions on therapeutic application of these compounds can be drawn. The Authors should add some self-criticism to the discussion pointing that these results are preliminary and there is still no data about potential side effects or (what is even more important) bioavailability of antcins.

I cannot find the procedure of the incubation of the cells with the antcins in Material and Methods section. It is the most important experimental part of this manuscript and it should be clearly described in a special subchapter. How the antcins were dissolved, in what solvent – such information should be given in detail because it is a constant problem with research on phytochemicals which are not dissolved in water.

Antcins are steroids, and the Authors should be more careful with their classification to “triterpenoids”. According to the present knowledge, steroids and triterpenoids should be treated separately, due to the differences in biosynthesis (squalene is a common precursor but afterwards the pathways are distinctly separated) and function. Therefore I do not agree with the statement in line 88 (“antcins are the typical triterpenoids, also classified as phytosterol”). In plus, mushrooms are not plants, so why suddenly “phytosterol”? Please change this fragment of the text. Fortunately it does not mean that the examples of bioactivities of “real” triterpenoids (e.g. ursolic or oleanolic acids, betulin etc.) could not be used in this manuscript, these compounds have similar polarity and due to their common origin can be described together in the context of potential therapeutic applications.

The style of writing is fully understandable, however, not always clear and I would suggest the careful reading (with “fresh eyes”) and some small improvements, particularly the sentences that are not correct grammatically or logically, e.g. line 85. “Additionally, ACE2 inhibition from plant extracts belongs to several phytochemical classes”.. (I guess, rather inhibitors?).

Reviewer 3 Report

It is an interesting study, but there are some concerns as follows.

L19: It was recently identified that Angiotensin Converting Enzyme 2 (ACE2); a receptor in the host epithelial cells required for SARS-CoV-2 cell entry and infection. => grammatically incorrect.

L45: However, despite insufficient evidence on the benefits and risks of Remdesivir, the World Health Organization (WHO) was recommended suspending the use of Remdesivir in COVID-19 patients [2]. Although, the number of positive cases and deaths due to COVID-19 continues to increase due to the unavailability of effective drugs to prevent disease or slow down the infection process. => grammatically incorrect.

L63-39: Please do not use bold letters.

L70: However, most studies have used in silico simulation models to identify potential agents, while few studies were conducted with in vitro or in vivo models. => Is it true? In any case, please show the references.

L73: cultured human colon epithelial cells in vitro => Please describe why you selected this cell line as well as why you did not use the in vivo models.

L74-94: These contents can be moved to the Introduction section.

L79: You can delete "Since".

L104: Studies were pointing out that gastrointestinal tissues and colon cell lines can be a platform to study the ACE2 activity. => There are no references.

L115: cell viability, however => cell viability; however

L183: Although, => Therefore,

L234: Please insert a space before "The percentage of ACE2 activity".

Round 2

Reviewer 2 Report

The manuscript has been improved according to my suggestions.

This manuscript is a resubmission of an earlier submission. The following is a list of the peer review reports and author responses from that submission.